# Peroxisomal Hydrogen Peroxide Metabolism and Signaling in Health and Disease

**DOI:** 10.3390/ijms20153673

**Published:** 2019-07-26

**Authors:** Celien Lismont, Iulia Revenco, Marc Fransen

**Affiliations:** Laboratory of Lipid Biochemistry and Protein Interactions, Department of Cellular and Molecular Medicine, KU Leuven—University of Leuven, 3000 Leuven, Belgium

**Keywords:** peroxisomes, flavin oxidases, catalase, hydrogen peroxide, cysteine oxidation, redox signaling, oxidative stress, organelle dysfunction, human disease

## Abstract

Hydrogen peroxide (H_2_O_2_), a non-radical reactive oxygen species generated during many (patho)physiological conditions, is currently universally recognized as an important mediator of redox-regulated processes. Depending on its spatiotemporal accumulation profile, this molecule may act as a signaling messenger or cause oxidative damage. The focus of this review is to comprehensively evaluate the evidence that peroxisomes, organelles best known for their role in cellular lipid metabolism, also serve as hubs in the H_2_O_2_ signaling network. We first briefly introduce the basic concepts of how H_2_O_2_ can drive cellular signaling events. Next, we outline the peroxisomal enzyme systems involved in H_2_O_2_ metabolism in mammals and reflect on how this oxidant can permeate across the organellar membrane. In addition, we provide an up-to-date overview of molecular targets and biological processes that can be affected by changes in peroxisomal H_2_O_2_ metabolism. Where possible, emphasis is placed on the molecular mechanisms and factors involved. From the data presented, it is clear that there are still numerous gaps in our knowledge. Therefore, gaining more insight into how peroxisomes are integrated in the cellular H_2_O_2_ signaling network is of key importance to unravel the precise role of peroxisomal H_2_O_2_ production and scavenging in normal and pathological conditions.

## 1. Introduction

Hydrogen peroxide (H_2_O_2_), the non-radical 2-electron reduction product of oxygen, is a natural metabolite commonly found in aerobic organisms [1]. For a long time, this compound was considered as an unwanted and rather detrimental by-product of oxidative metabolism [2]. However, during the last decades, H_2_O_2_ has moved into the forefront as a central redox signaling molecule in many biological processes such as cell proliferation and differentiation, tissue repair, inflammation, circadian rhythm, and aging [1,3]. The signaling properties of H_2_O_2_ can be attributed to its relative stability, diffusibility, and selective reactivity (see Section 2). However, whether H_2_O_2_ acts as a signaling molecule or leads to oxidative damage of biomolecules, a condition denoted as oxidative stress, depends on the cellular context, its local concentration, and the kinetics of its production and elimination [4]. For example, in the presence of free redox-active metal ions, non-toxic physiological levels of H_2_O_2_ can give rise to highly toxic hydroxyl radicals (^●^OH), which indiscriminately oxidize virtually any organic molecule they encounter [5,6].

Under steady state conditions, the in cellulo production and consumption of H_2_O_2_ are balanced. Major metabolic sources of H_2_O_2_ include the flavin-dependent oxidases (e.g., the endoplasmic reticulum oxidoreductase 1 (ERO1) [7]; the acyl-coenzyme A (acyl-CoA) oxidases in peroxisomes (see Section 3.1)); and superoxide dismutases (SODs) [8]. The latter group of enzymes catalyzes the dismutation of superoxide anion radicals (O_2_^●−^), which are predominantly produced by the mitochondrial electron transport chain [9] and membrane-associated NADPH oxidases (NOXs) that are located in various subcellular compartments [10]. Major H_2_O_2_-scavenging enzymes include catalase (CAT) and various thiol-based peroxidases such as glutathione peroxidases (GPXs) and peroxiredoxins (PRDXs) [11]. While CAT is predominantly located in peroxisomes (see Section 3.2), GPXs [12] and PRDXs [13] are often localized in various subcellular compartments.

Peroxisomes are cell organelles that are best-known for their involvement in cellular lipid metabolism [14]. In mammals, this entails the α- and β-oxidation of fatty acids and the biosynthesis of ether-phospholipids, bile acids, and docosahexaenoic acid [15]. Other metabolic functions of peroxisomes in mammals include glyoxylate detoxification, amino acid catabolism, polyamine oxidation, and the production and scavenging of reactive oxygen and nitrogen species (ROS and RNS, respectively) [16,17]. In addition, these organelles are also increasingly recognized as important hubs in innate immune-, lipid-, inflammatory-, and redox-signaling networks [18]. To perform these functions, peroxisomes can dynamically regulate their number, shape, and protein content in response to changing environmental conditions [19]. They also have to stay in close communication with other subcellular compartments [20]. The formation and maintenance of peroxisomes requires a specialized set of proteins, called peroxins (acronym: PEX; the number refers to their order of discovery) [21], and superfluous and dysfunctional organelles are targeted for lysosomal degradation through a process known as pexophagy [22]. Alterations in any of these metabolic, signaling, or other pathways have been linked to multiple genetic (e.g., Zellweger syndrome) [23], infectious [24], and oxidative stress-related (e.g., neurodegeneration, diabetes, and cancer) diseases [25,26]. In the following sections, we focus on the role of mammalian peroxisomes in cellular H_2_O_2_ metabolism and how perturbations in this process may affect cellular function and organismal health.

## 2. The Basic Concepts of H_2_O_2_ Signaling

Like many other signaling pathways, hallmarks of H_2_O_2_ signaling include messenger formation, messenger metabolism, messenger action, and recovery [27]. Given its relative stability (cellular half-life: ~1 ms), diffusibility, and selective reactivity, H_2_O_2_ is often put forth as the most important redox signaling molecule [1,28,29]. This non-radical ROS can be formed and degraded in different subcellular compartments (see Section 1), and its local concentration depends on its rates of synthesis, degradation, and diffusion (Figure 1). Importantly, although H_2_O_2_ is a non-charged molecule, its dipole moment is higher than that of water, thereby severely limiting its transport across lipid membranes by simple diffusion [29]. Indeed, efficient permeation of this molecule across biological membranes requires specific channel proteins, referred to as peroxiporins [1]. The spatial separation of sources and sinks as well as its diffusion away from the site of production and across biomembranes lead to the formation of intracellular H_2_O_2_ gradients, which determine the behavior of responsive systems [6].

The primary messenger action of H_2_O_2_ depends on its ability to oxidize a variety of target proteins with a high degree of specificity, predominantly through reaction with nucleophilic cysteine thiolate groups (Cys-S^−^) that can be found in specific protein microenvironments [1]. Oxidation of such deprotonated cysteine residues can lead to the formation of unstable sulfenic acid (-SOH) intermediates that can (i) be reduced again, (ii) react with other proximal thiol groups to form intra- or intermolecular disulfide bonds, or (iii) undergo hyperoxidation to form sulfinic (-SO_2_H) and then sulfonic (-SO_3_H) acids [11,30]. Disulfide bond formation can induce conformational changes, leading to alterations in macromolecular interactions, protein localization, function, activity, and/or stability; and protein disulfides can be converted back to their reduced state by components of the glutaredoxin (GLRX)/glutathione (GSH)/glutathione-disulfide reductase (GSR)- and thioredoxin (TXN)/thioredoxin reductase (TXNRD)-containing antioxidant systems (Figure 1) [31,32]. As such, oxidation-susceptible cysteine thiols can function as regulatory switches that transmit information along a signaling cascade after being oxidized by H_2_O_2_ [11].

Importantly, depending on the context and circumstances (e.g., the subcellular location, the H_2_O_2_ concentration, and duration of exposure, etc.), the oxidation of signaling proteins may occur through direct reaction of H_2_O_2_ with hyper-reactive thiols or indirectly through thioredoxin- or peroxiredoxin- catalyzed redox relay reactions (Figure 1) [33,34]. This implies that the thiol peroxidases involved may display a dual function: on one hand, they can function as H_2_O_2_ scavengers that counteract redox signaling; on the other hand, they may act as enablers of protein thiol oxidation by transferring oxidative equivalents from H_2_O_2_ to redox-regulated proteins [34]. In the latter mechanism, the target specificity is likely to be determined by protein–protein interactions. In this context, it is relevant to mention that the reducing power required to maintain thiol redox networks is provided by NADPH [35].

To illustrate the complexity of H_2_O_2_-mediated signaling; we briefly focus on one specific example: the activity modulation of nuclear factor erythroid 2-related factor 2 (NFE2L2). NFE2L2, also known as NRF2, is a transcription factor regulating the expression of genes containing an antioxidant/electrophile response element motif in their promoter [36]. Under homeostatic conditions, NFE2L2 is constitutively degraded by different pathways: on one hand, the protein interacts with Kelch-like ECH-associated protein 1 (KEAP1), a redox-regulated substrate adapter protein of the cullin 3-dependent E3 ubiquitin ligase complex that mediates the ubiquitination and subsequent proteasomal degradation of KEAP1-associated proteins; on the other hand, the protein can be phosphorylated by glycogen synthase kinase 3β (GSK-3β), thereby creating a phosphodegron that drives protein degradation through interaction with a ubiquitin ligase [36,37]. However, upon exposure to H_2_O_2_, specific cysteine residues of KEAP1 (e.g., Cys151, Cys171, Cys273, and Cys288 in the human protein) are oxidized, thereby inducing conformational changes interfering with the ability of KEAP1 to ubiquitinate NFE2L2 [38]. In addition, given that (i) phosphorylation of GSK-3β at serine 9 inhibits the kinase’s activity [39], and (ii) multiple kinases (e.g., the serine/threonine protein kinases B and C, the mitogen-activated protein kinases 1 and 14) that can mediate this phosphorylation event are activated by H_2_O_2_ [36], such condition also results in a KEAP1-independent activation of NFE2L2. Importantly, electrophiles such as H_2_O_2_ can also modulate the activity of NFE2L2 at other levels. For example, they can (i) enhance the translation rate of the NFE2L2 mRNA [40,41], (ii) stimulate the nuclear accumulation of the NFE2L2 by freeing a nuclear localization sequence that is not surface-exposed in the KEAP1-bound transcription factor and/or by oxidatively modifying (or masking) nuclear export sequences in KEAP1 or the NFE2L2–KEAP1 complex [36], (iii) increase the acetylation state of NFE2L2, a condition reported to correlate with DNA binding and transcription factor activity [42,43], and (iv) affect the interaction of NFE2L2 with its transcriptional regulators [36].

## 3. Players in Peroxisomal H_2_O_2_ Metabolism and Transport

Currently, it is clear that peroxisomes contain both enzymes that can produce (see Section 3.1) or scavenge (see Section 3.2) H_2_O_2_. In addition, there is compelling evidence that the peroxisomal membrane contains non-selective pore-forming proteins that allow the exchange of small metabolites (see Section 3.3). As such, these organelles can be expected to influence local H_2_O_2_ gradients and, therefore, H_2_O_2_-mediated signaling events. Whether peroxisomes act as a net sinks or sources of H_2_O_2_ (Figure 2) most likely depends on the cell type, its physiological state, and the microenvironment.

### 3.1. H_2_O_2_-Generating Systems

Peroxisomes are metabolically active organelles involved in a variety of biochemical processes (see Section 1). A specific subset of enzymes acting in these processes include the flavin adenine dinucleotide (FAD)- or flavin mononucleotide (FMN)-dependent oxidases (for a detailed list, see [44]), of which the reduced flavin forms are regenerated by reduction of molecular oxygen (O_2_) to H_2_O_2_. In this context, it is important to note that (i) unlike mitochondria, peroxisomes lack a respiratory chain [45], and (ii) depending on the oxidase, the cofactor can be loosely or firmly bound [46,47]. Natural substrates of the mammalian H_2_O_2_-producing peroxisomal oxidases include, among others, different types of fatty acids (e.g., very-long-chain fatty acids, 2-methyl-branched fatty acids, 2-hydroxy fatty acids, and bile acid intermediates) [48], d-amino acids (e.g., d-Ser) [49], polyamines (e.g., *N1*-acetylspermine) [50], glycolate [51], and pipecolic acid [52]. Depending on the activity of these oxidases, which may be regulated by numerous posttranslational modifications (e.g., phosphorylation, acetylation, ubiquitination, succinylation, mono- or di-methylation [53]), and the intraperoxisomal H_2_O_2_ scavenging rates (see Section 3.2), changes in substrate availability or enzyme activity may result in altered H_2_O_2_ levels. This is nicely illustrated by a recent study showing that (i) lysine succinylation stimulates the activity of acyl-CoA oxidase 1 (ACOX1), a rate-limiting enzyme in peroxisomal fatty acid β-oxidation, (ii) sirtuin 5 (SIRT5), an NAD-dependent protein lysine desuccinylase, can locate to peroxisomes where it binds to and desuccinylates ACOX1, and (iii) downregulation of SIRT5 increases ACOX1 activity, peroxisomal H_2_O_2_ production, and oxidative DNA damage in cultured HepG2 liver cells, mouse livers, and/or human hepatocellular carcinoma samples [54].

### 3.2. H_2_O_2_-Elimination Systems

Currently, it is clear that mammalian peroxisomes are equipped with at least two H_2_O_2_-eliminating enzyme systems, of which the best characterized enzyme is CAT. Depending on the H_2_O_2_ concentration and the presence of other metabolic hydrogen donors (AH_2_) such as short-chain aliphatic alcohols, formate, or nitrite, this heme-containing protein can scavenge H_2_O_2_ in a catalatic (2 H_2_O_2_ → 2 H_2_O + O_2_) or peroxidatic (H_2_O_2_ + AH_2_ → A + 2 H_2_O) manner [55]. In contrast to other H_2_O_2_-decomposing enzymes, these reactions occur without the use of reducing equivalents [56]. CAT is an abundant, predominantly peroxisomal enzyme. However, depending on the cell type and environmental conditions, the protein may also be (partially) localized to the cytosol and nucleus [57,58]. In this context, it is worth noting that oxidative stress impairs the import efficiency of CAT into peroxisomes, a phenomenon that apparently protects the cytosol against H_2_O_2_-induced insults [59]. In addition, upon exposure of mammalian cells to external H_2_O_2_ or other ROS stimuli, (cytosolic) catalase can be phosphorylated by protein kinase C delta (PRKCD) at Ser167 [60] and by the Abelson tyrosin-protein kinases ABL1 and ABL2 at both Tyr231 and Tyr386 [61], thereby enhancing its activity. On the other hand, the activity of CAT can be down-regulated through nitrosylation of Cys377 [62] and *S*-thiolation at not yet identified cysteine residues [63,64]. Finally, CAT can also be phosphorylated [65], acetylated [66], succinylated [54,67], monomethylated [68,69], ubiquitinated [70,71], and sumoylated [72] on many other residues. However, how these posttranslational modification events affect CAT localization and/or activity remains to be determined. From these observations, combined with the fact that CAT is not essential for life [73,74], it is apparent that more studies are needed to fully understand the precise regulation and physiological function of this H_2_O_2_-scavenging enzyme in peroxisomal redox biology.

Another sink for H_2_O_2_ inside peroxisomes is PRDX5, a thiol-dependent monomeric peroxidase that is also located in the cytosol, the nucleus, and mitochondria [75]. PRDX5 can also reduce peroxynitrite (ONOO^−^) and a variety of lipid peroxides (LOOH) [76]. Currently, it is widely accepted that PRDX5 uses an NADPH-dependent thioredoxin (TXN)/TXN reductase (TXNTR) system to reduce its substrates [31]. Intriguingly, no such enzyme system has yet been identified in mammalian peroxisomes. Like CAT, PRDX5 can also be phosphorylated [77,78], acetylated [66,79], succinylated [67], glutathionylated [80], and ubiquitinated [70,71]. In addition, the protein can undergo disulfide bond formation [81,82]. Once again, how these posttranslational modifications affect PRDX5 activity remains to be clarified. The physiological function of peroxisomally located PRDX5 is also not yet clear. On one hand, the protein may act as an antioxidant by inactivating H_2_O_2_ before it can modify redox-sensitive cysteines [83]. On the other hand, it may act as a redox relay factor by transferring oxidizing equivalents from H_2_O_2_ to target proteins through thiol-disulfide reshuffling [33,34]. In the latter case, the peroxisomal pools of PRDX5 and CAT may play non-overlapping roles in H_2_O_2_ clearance, a paradigm supported by the observation that both antioxidant enzymes present distinct kinetic characteristics. Indeed, PRDX5 and CAT scavenge H_2_O_2_ in the low micromolar and low millimolar range, respectively, and the maximum rate of H_2_O_2_ removal is orders of magnitude more for CAT than for PRDX5 [76,84].

Besides CAT and PRDX5, H_2_O_2_ may also be removed by glutathione peroxidases (GPXs), a family of antioxidant enzymes that typically use GSH as reductant [85]. However, despite the fact that rat liver peroxisomes appear to contain GPX activity [86], no such enzyme has yet been identified at the protein level.

### 3.3. H_2_O_2_ Permeation across the Peroxisomal Membrane

Throughout the years, it has become clear the biological membranes, including the peroxisomal one, act as permeability barriers for H_2_O_2_ [87,88]. Importantly, to serve as a subcellular platform for H_2_O_2_ signaling, it is essential that peroxisomes can exchange this redox messenger with their environment. That this is indeed the case, has already been demonstrated in vitro [89,90], in cellulo [91], and in slices of liver, lung, and lenses from CAT-deficient mice [92]. In general, it is thought that this permeation process is governed by peroxiporins (see Section 2), a class of proteins that has not yet been assigned to peroxisomes. However, here it is important to note that the peroxisomal membrane contains a non-selective pore-forming protein, termed PXMP2, that allows free diffusion of small molecules (< 300–600 Da) [93]. Nonetheless, we have recently demonstrated that neither this peroxisomal membrane protein (PMP) nor PEX11B, another widely expressed PMP whose yeast homologue enables the permeation of molecules up to 400 Da [94,95], are essential for the transport of H_2_O_2_ across the peroxisomal membrane [96].

## 4. The Emerging Roles of Peroxisomes in Cellular H_2_O_2_ Signaling

A general requisite for signal propagation is the spatial segregation of opposing reactions, a condition resulting in the formation of concentration gradients [97]. Therefore, in order to function as a signaling messenger, H_2_O_2_ does not only need to be produced, but also to be enzymatically degraded or removed. This degradation/removal process is essential to reduce the refractory time, which is defined as the recovery period necessary for detecting successive signals [98]. Otherwise, H_2_O_2_ levels will build up, thereby preventing further signaling. Over the years, it has become clear that changes in peroxisomal H_2_O_2_ production or CAT activity can modulate the cellular thiol-disulfide state [99]. In the following sections, we first outline what is known about the molecular targets of peroxisome-derived H_2_O_2_ and how changes in CAT activity alter cellular thiol-disulfide homeostasis (see Section 4.1). Next, we discuss how peroxisomes may act as modulators of diverse biological processes regulated by H_2_O_2_ (see Section 4.2). Finally, we provide a brief overview of how imbalances in any of these processes may contribute to disease (see Section 4.3).

### 4.1. Molecular Targets

As outlined above (see Section 2), protein cysteinyl residues are the prime mediators of H_2_O_2_ signaling [100]. Unfortunately, until now, no global proteomics data are available for redox-active thiols that can be modified by peroxisome-derived H_2_O_2_. To address this gap, we recently developed a human HEK-293 cell line that can be used to selectively induce H_2_O_2_ production inside peroxisomes in a time- and dose-controlled manner [101], and studies to inventory the peroxisomal H_2_O_2_-dependent sulfenome are ongoing. However, during the validation of this cell model, we could provide evidence that peroxisome-derived H_2_O_2_ can oxidize redox-sensitive cysteine thiols in proteins within and outside the peroxisomal compartment. Specific examples include the forkhead box O3 transcription factor FOXO3, the p50 and p65 subunits of the transcription factor nuclear factor kappa B (NF-κB), the tumor suppressor phosphatase PTEN, the peroxisomal import receptor PEX5, and the antioxidant enzyme PRDX5 [101].

An alternative approach to study the role of peroxisomal H_2_O_2_ metabolism in cellular thiol-disulfide homeostasis is to interfere with CAT activity. Here it is interesting to mention that (i) CAT activity, and not the cellular glutathione levels, appear to dominate the resistance of cells to ROS [102], (ii) treatment of Chang liver cells with 3-amino-1,2,4-triazole, an irreversible inhibitor of CAT activity, increases the protein disulfide levels by 20% [103], and (iii) cardiac-specific overexpression of CAT in mice decreases oxidative cysteine modification of cardiac proteins [104]. Whether the observed changes in cysteine oxidation are caused by alterations in the release of peroxisome-derived H_2_O_2_ or by changes in the capacity of peroxisomes to scavenge extra-peroxisomal H_2_O_2_ (Figure 2), remains to be determined.

### 4.2. Biological Processes

Despite the limited number of protein targets that have been identified for peroxisome-derived H_2_O_2_ (see Section 4.1), an extensive literature exists on the potential involvement of peroxisomes in the H_2_O_2_-mediated regulation of various fundamental biological processes. The aim of this Section is not to provide an exhaustive overview of this subject, but to outline some relevant examples (Figure 3).

#### 4.2.1. Gene Expression

The synthesis, stability, subcellular localization, and/or activity of many transcription factors are regulated by H_2_O_2_ [36]. Currently, there is strong evidence that also peroxisomal H_2_O_2_ metabolism may contribute to these processes, thereby modulating gene expression. This is perhaps best illustrated by the observation that whole-genome expression profiling studies have revealed that CAT activity modulates the expression of numerous genes, both in human cells [105] and in mice [106]. Another example is that, depending on the cell type and experimental conditions [107], sustained production of peroxisomal H_2_O_2_ [108,109,110], CAT inhibition [111], or CAT overexpression [112] can activate [108,109,110] or inhibit [111,112] the activity of NF-κB. These and other findings provide empirical evidence that alterations in peroxisomal H_2_O_2_ metabolism can modulate gene expression (Figure 3). However, the biological significance and underlying mechanisms of these observations remain largely to be established.

#### 4.2.2. Cell Fate Regulation

Cell fate decisions such as growth, proliferation, differentiation, senescence, and apoptosis are impacted by multiple environmental and biological cues, including H_2_O_2_ [113]. This molecule can instruct such decisions, directly or indirectly, by affecting the functionality of transcription factors and/or other proteins (e.g., kinases, phosphatases, proteases, antioxidant enzymes, etc.) involved in key signal transduction pathways [114]. Importantly, changes in peroxisomal H_2_O_2_ metabolism have been reported to influence cell fate transitions [99]. However, although these transition switches can be expected to occur through modification of the epigenetic landscape [115] and transcriptional responses (see Section 4.2.1), the driving mechanisms remain largely to be established.

Given that low levels of H_2_O_2_ can promote cell proliferation and differentiation [113], it is not surprising that CAT overexpression has been shown to reduce the growth of various cell types (e.g., rat aortic smooth muscle cells [116], human aortic endothelial cells [117], human MCF-7 breast cancer cells [118], A-375 amelanotic melanoma cells [105], and human promyelocytic HL-60 cells [119]). Interestingly, such treatment has been reported to delay the resting (G0)/gap phase 1 (G1) to synthesis (S)-phase transition in mouse aortic endothelial cells during cell cycle progression [120]. In addition, CAT overexpression in HL-60 (or the human promonocytic cell line U-937) and A-375 cells has been demonstrated to potentiate macrophage and melanocyte differentiation, respectively [105,121].

Depending on the cell type and context, disturbances in peroxisomal H_2_O_2_ metabolism may also drive cell transformation (see Section 4.3.5), promote cellular aging and senescence, or trigger cell death. Indeed, chronic reduction of CAT activity in human cells has been documented to increase oxidative damage, enhance the secretion of matrix metalloproteinases, and impair mitochondrial function (see Section 4.2.3) (Figure 3) [122]. In addition, there is strong evidence that CAT can act as a protectant against apoptosis induced by multiple types of oxidative insults (e.g., ultraviolet and ionizing radiation [123,124], arsenic trioxide treatment [125], or P53-induced oxidative stress [126]). Also, peroxisomally located PRDX5 has been shown to have a cytoprotective effect against H_2_O_2_-induced cytotoxicity [127]. Importantly, overexpression of CAT may also sensitize cells (e.g., mouse hepatocytes and fibroblasts [128], human alveolar macrophages [129], and MCF-7 cells [118]) as well as animals (e.g., non-obese diabetic mice [130]) to different types of stressors (e.g., paraquat [128], tumor necrosis factor-alpha (TNFα) [128], asbestos [129], cyclophosphamide [130], paclitaxel [118], etoposide [118], and arsenic trioxide [118]). The latter findings clearly demonstrate that high CAT activity can dampen H_2_O_2_ signaling. Finally, it has been reported that the lipotoxicity of saturated non-esterified fatty acids in rat insulin-producing cells, which are catalase-poor, is caused by excessive production of H_2_O_2_ via peroxisomal β-oxidation [131].

#### 4.2.3. Mitochondrial Function

Peroxisomes and mitochondria cooperate in various metabolic (e.g., β-oxidation of fatty acids, ROS metabolism) and signaling (e.g., antiviral innate immune signaling) pathways [48,132], and disturbances in any of the peroxisomal processes—including H_2_O_2_ metabolism—can result in reduced mitochondrial fitness [133,134]. For instance, several studies have documented that inhibition of CAT activity rapidly increases mitochondrial ROS levels [135,136] and impairs mitochondrial membrane potential and aconitase activity [122,137]. In addition, CAT overexpression has been reported to safeguard mitochondrial fitness, thereby protecting the cells against stress insults [122,136,138]. Also here, whether this redox interplay between peroxisomes and mitochondria is a direct result of alterations in the release of peroxisome-derived H_2_O_2_ or caused by changes in the capacity of peroxisomes to scavenge extra-peroxisomal H_2_O_2_, remains to be investigated. Indeed, to the best of our knowledge, evidence demonstrating that peroxisome-derived H_2_O_2_ can directly impact mitochondrial function is still lacking.

### 4.3. Diseases

Peroxisomes have the intrinsic potential to mediate (see Section 3.1 and Section 3.3) and modulate (see Section 3.2 and Section 3.3) H_2_O_2_-driven signaling events. As such, it may not come as a surprise that imbalances in peroxisomal H_2_O_2_ metabolism have been associated with multiple oxidative stress-related disease states, including obesity, diabetes, ischemia reperfusion, noise-induced hearing loss, neurodegeneration, aging, and tumor initiation and progression. For a detailed overview on this topic, we refer the reader to another recent review [99]. However, to illustrate the concept, we focus here on some other examples that can all be linked to alterations in CAT activity (Figure 3). Note that the role of this enzyme in protecting cells and tissues against H_2_O_2_-induced injury has already been amply documented (see below). In addition, from these and other findings, it is clear that inherited catalase deficiencies should be considered as a strong risk factor for aging-related pathological changes [74].

#### 4.3.1. Heart Disease

Transgenic overexpression of CAT in mouse hearts has been reported (i) to prevent adverse myocardial remodeling and progression to overt heart failure in a mouse model of dilated cardiomyopathy [139], (ii) to protect the heart from (post-)ischemia-reperfusion injury [140,141], diabetic cardiomyopathy and dysfunction [112,142], and lipopolysaccharide-induced cardiac dysfunction and mortality [143], (iii) to preserve cardiac function after myocardial infarction, at least at later time points [144], and (iv) to alleviate cardiac diseases and aging [104,145]. These improved phenotypes have been linked to an overall decrease in oxidative stress [142,143], a shift of the protein thiol/disulfide balance towards thiols [104], a decrease in endothelial nitric oxide synthase activity [142], less nitration of key enzymes involved in energy metabolism [112], a decline in NF-κB signaling [112], a reduction in proinflammatory cytokine release [144], and a marked protection against myocyte hypertrophy, myocyte apoptosis, and interstitial fibrosis [139].

#### 4.3.2. Kidney Disease

CAT overexpression has also been demonstrated to provide renoprotection. Indeed, overexpression of this enzyme in renal proximal tubular cells of angiotensinogen transgenic mice [146], type 1 diabetic Akita transgenic mice [147,148], or type 2 diabetic *db*/*db* mice [149] has been shown to mitigate oxidative stress and prevent hypertension, albuminuria, tubulointerstitial fibrosis, and tubular apoptosis [146,147,148]. At the molecular level, these phenomena have been associated with attenuated angiotensinogen and proapoptotic gene expression [149,150].

#### 4.3.3. Insulin Resistance and Diabetes

Pancreatic β-cells are very vulnerable to oxidative stress, a phenomenon of importance in type 1 diabetes and islet transplantation [151]. Interestingly, overexpression of CAT in murine pancreatic β-cells has been demonstrated to have no detrimental effects and to provide marked protection of islet insulin secretion against H_2_O_2_- and streptozocin-mediated β-cell dysfunction [152]. In addition, it has been reported that overexpression of CAT improves mitochondrial function in insulin resistant muscle cells, thereby enhancing glucose and fatty acid metabolism [138]. This finding is in line with other studies showing that (i) CAT deletion exacerbates the pre-diabetic phenotype in mice [106], and (ii) overexpression of CAT in obese mice has a positive influence on energy expenditure and metabolic parameters such as leptin and adiponectin levels [153].

#### 4.3.4. Cardiovascular Disease

Oxidative stress is a potent contributing factor to cardiovascular disease [154], and overexpression of CAT in vascular [155,156] or aortic [157] smooth muscle cells has been shown (i) to decrease oxidized lipid-induced cytotoxicity in vitro [155], and (ii) to prevent pathological mechanical changes underlying abdominal aneurysm formation in transgenic mice, primarily through modulation of matrix metalloproteinase activity [156,157]. However, transgenic mice with specific overexpression of CAT in myeloid lineage cells display impaired post-ischemic neovascularization, a phenotype associated with a blunted inflammatory response (e.g., lower levels of inflammatory markers; reduced macrophage infiltration) in ischemic tissues [158].

#### 4.3.5. Cancer

Compared to normal cells, cancer cells frequently produce elevated levels of ROS compared to their normal counterparts [159]. These molecules can act as pro-tumorigenic signals that promote, among others, abnormal cell growth, migration, resistance to apoptosis, adaptations to hypoxia, and genetic instability [159]. Interestingly, low CAT activity has been associated with an increased risk factor for many different cancers, including skin cancer [160], colorectal cancer [161], breast cancer [162], invasive cervical cancer [163], ovarian cancer [164], and prostate cancer [165,166,167]. On the other hand, overexpression of CAT in MCF-7 mammary cancer cells has been reported to result in a less aggressive phenotype and an altered response to chemotherapy [118]. Finally, CAT activity has also been linked to the cathepsin-induced migration and invasion of human lung cancer cells [168].

#### 4.3.6. Neurodegenerative Disease

Oxidative stress is a common denominator of various neurodegenerative disorders, including Alzheimer’s disease, Parkinson’s disease, and amyotrophic lateral sclerosis [169]. The underlying reason is that the brain is exquisitely vulnerable to oxidative damage because of its high oxygen consumption, elevated concentrations of unsaturated lipids, and modest antioxidant defense compared to other tissues [169,170]. Interestingly, patients suffering from a peroxisomal deficiency also typically develop severe neurological deficits [171]. However, at the moment, little is known about how alterations in peroxisomal H_2_O_2_ metabolism contribute to brain homeostasis and health. To the best of our knowledge, there are no reports that comprehensively study how selective alterations in peroxisomal H_2_O_2_ production influence physiological and pathological brain processes. Nonetheless, there are a number of in vitro and in vivo studies that investigate the role of CAT as neuroprotective agent. For example, it has been demonstrated that increased levels of (peroxisome-targeted) CAT (i) protect cultured primary human neurons from H_2_O_2_-mediated cytotoxicity [172], (ii) safeguard cultured SH-SY5Y human neuroblastoma cells from β-amyloid-induced oxidative stress [173], (iii) reduce the toxicity of amyloid-β_25–35_ in rat brain [174], and (iv) protect isolated rat brain mitochondria against the toxic effects of 6-hydroxydopamine on mitochondrial respiration [175]. In addition, there is compelling evidence that binding of the neurotoxic β-amyloid peptide to CAT decreases its activity [176], a phenomenon that may explain why CAT activity is reduced in the brain of Alzheimer’s disease patients [177]. Nevertheless, despite these observations, other studies do not support the idea of CAT being a susceptibility factor for Alzheimer’s disease [178], Parkinson’s disease [179,180], or familial amyotrophic lateral sclerosis [181].

## 5. Conclusions, Challenges, and Perspectives

As reviewed here, there is currently overwhelming evidence supporting the view that peroxisomes have the intrinsic ability to mediate and modulate H_2_O_2_-driven biological processes. In addition, there is growing consensus that perturbations in peroxisomal H_2_O_2_ metabolism can elicit adaptive or maladaptive responses that mitigate or aggravate the impact of the underlying cause. However, the specific mechanisms and physiological consequences of these events remain largely to be explored. The most critical questions that need to be answered include (i) the identification and functional dissection of redox-sensitive proteins that can be reversibly oxidized by peroxisome-derived H_2_O_2_, (ii) the nature of the proteins involved in the transport of H_2_O_2_ and other relevant redox species across the peroxisomal membrane, and (iii) the biological consequences of changes in peroxisomal H_2_O_2_ metabolism on cellular signaling networks that drive physiological or pathological responses. These questions are further elaborated in the next sections.

As mentioned above (see Section 4.1), there are presently no mammalian proteomics data for redox-active thiols that can be modified by peroxisome-derived H_2_O_2_. The main underlying reason can be attributed to the fact that, until recently, an experimental model to selectively produce physiological concentrations of H_2_O_2_ inside peroxisomes in a time- and dose-controlled manner was lacking. However, given that such a model [101] as well as a genetic tool to capture and affinity purify sulfenic acid-containing proteins [182] are currently available, proteome-wide unbiased identification of primary targets of peroxisome-derived H_2_O_2_ can be expected soon. In this context, it will also be interesting to see whether H_2_O_2_ generated inside peroxisomes or other locations (e.g., mitochondria or the cytosol) yields many specific or common targets.

The steady-state levels of H_2_O_2_ inside peroxisomes are determined by the rates of its synthesis, degradation, and diffusion. Besides catalase, also NAD(P)H- and GSH-powered redox systems can be expected to play a role in peroxisomal redox homeostasis (these systems form a complex network of interactions with GPXs, TXNs, and PRDXs) [30,183]. However, as for H_2_O_2_ (see Section 3.3), it is still unclear how GSH, GSSG, NAD(P)^+^, and NAD(P)H are transported across the peroxisomal membrane and how peroxisomes regulate their GSH/GSSG and NAD(P)^+^/NAD(P)H pools. In addition, the electron donor for peroxisomal PRDX5 remains to be identified. For more details on these topics, we refer the reader to other recent reviews [48,99].

A last unresolved but pertinent question is how alterations in peroxisomal H_2_O_2_ metabolism contribute to cellular and organismal physiology. Here, it is important to highlight that more research needs to be done to determine under which conditions peroxisomes serve as net sources or sinks for H_2_O_2_. In addition, despite the fact that the redox proteome can provide a link between metabolism and sulfur switch-controlled signaling events [183], it is important to realize that (i) the subgroup of H_2_O_2_-sensitive cysteine residues shows less conservation than their redox-insensitive counterparts, and (ii) the H_2_O_2_-dependent redoxome can vary dramatically between different cell types [100]. These factors complicate the interpretation of how peroxisome-derived H_2_O_2_ may modulate redox-driven intracellular signaling events and pose a significant challenge for translating the in cellulo data to in vivo models and subsequently to clinical practice.

Taken together, there is convincing evidence that peroxisomes do serve as an intracellular hub in H_2_O_2_ metabolism and signaling. However, additional work is needed to better understand how cells decode and integrate these cues to produce coherent responses. The outcome of such studies can be expected to advance redox medicine.

## Figures and Tables

**Figure 1 ijms-20-03673-f001:**
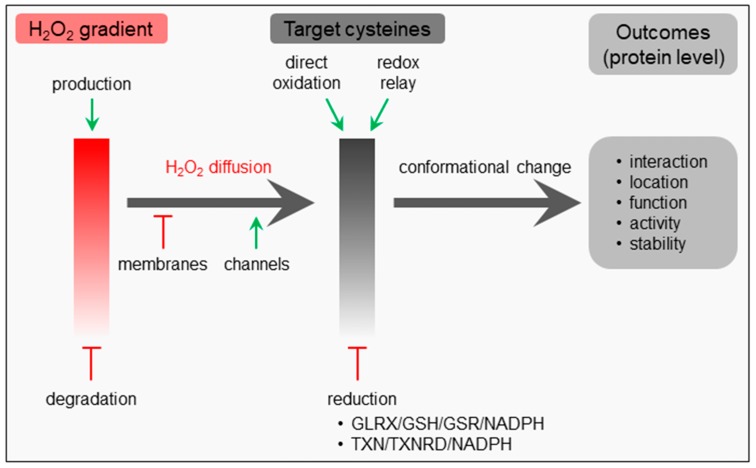
General principles of H_2_O_2_ signaling. GLRX, glutaredoxin; GSH, reduced glutathione; GSR, glutathione-disulfide reductase; NADPH, nicotinamide adenine dinucleotide phosphate (reduced); TXN, thioredoxin; TXNRD, thioredoxin reductase.

**Figure 2 ijms-20-03673-f002:**
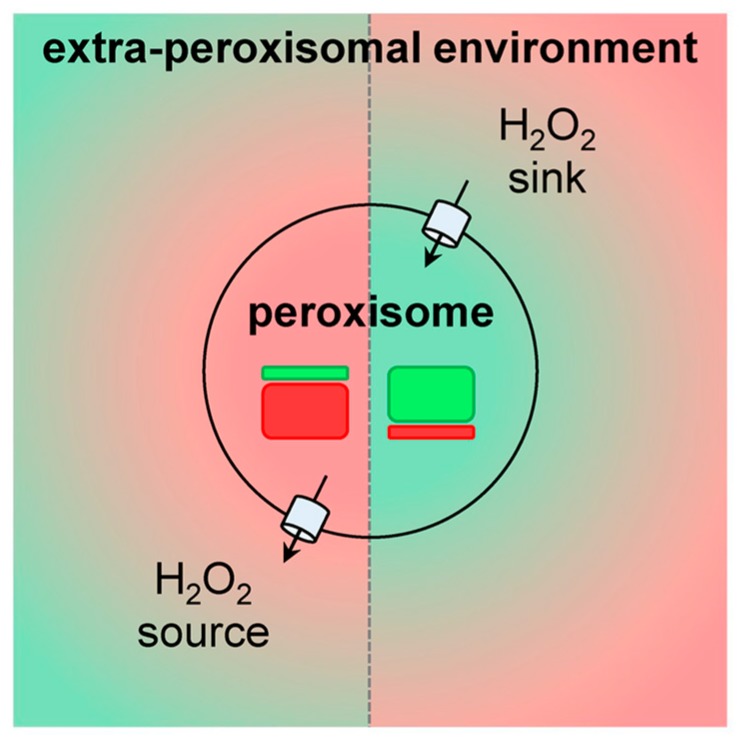
Peroxisomal hydrogen peroxide (H_2_O_2_) metabolism and its potential effects on intracellular H_2_O_2_ gradients. H_2_O_2_-producing and -degrading enzymes are depicted as red and green rectangles, respectively.

**Figure 3 ijms-20-03673-f003:**
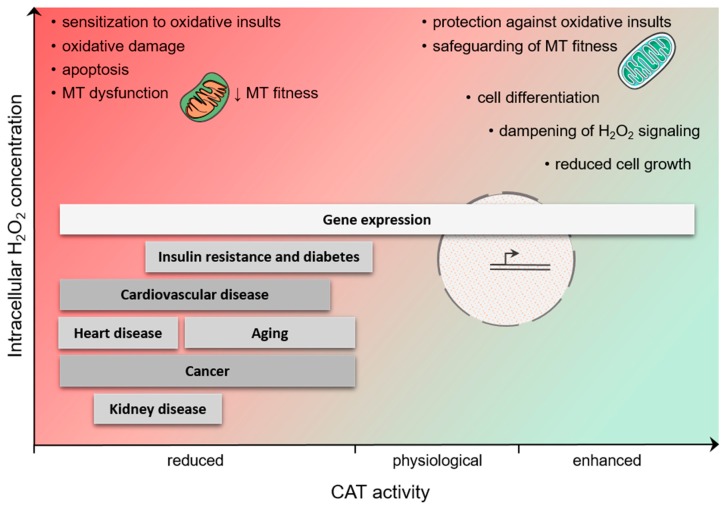
Schematic presentation of how alterations in catalase (CAT) activity may shape intracellular H_2_O_2_ gradients, thereby impacting multiple biological processes and contributing to disease initiation and progression. MT, mitochondria.

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
