# Peer review of "Peroxisomal Hydrogen Peroxide Metabolism and Signaling in Health and Disease"

_ijms, 2019, doi:10.3390/ijms20153673_

Round 1

Reviewer 1 Report

The Review by Lismont C et al. is very interesting, clear and well written. It gives a new point of view about the modulation of biological processes by H2O2 and suggests many questions that are yet unresolved. I recommend to accept the manuscript.

Only two minor points:

- pag 1, line 43: delete the second bracket after "see 3.1."

- pag 6, line 239: specify "human" before the word "cells [97]" 

Author Response

General comments – The reviewer states: ‘The Review by Lismont C et al. is very interesting, clear and well written. It gives a new point of view about the modulation of biological processes by H2O2 and suggests many questions that are yet unresolved. I recommend to accept the manuscript.’
We thank the reviewer for his/her positive assessment of our manuscript.

Comment 1 – The reviewer states: ‘page 1, line 43: delete the second bracket after "see 3.1."’

Given that the first and second closing parentheses after "see 3.1. " denote the end of the cross-reference to section 3.1. and the end of the parenthetical material, respectively, we are convinced that both brackets are required for syntax correctness. We hope this clarification resolves the confusion.

Comment 2 – The reviewer states: ‘page 6, line 239: specify "human" before the word "cells [97] and …’.

The suggested change has been made (see revised manuscript page 6, line 261).

Reviewer 2 Report

The review is very well written and documented. It provides a very interesting overview of peroxisomal hydrogen peroxide metabolism and signaling. I have however some minor criticisms to formulate.

Concerning the signaling role of hydrogne peroxide (section 2), I would suggest to mention a few articles dealing with H2O2-regulated transcription factors (NRF2…). The review of Marino et al (Redox Biology 2 :535-62, 2014) may help to build a short paragraph to complete the section. It may help to understand how peroxisomal metabolism can impact various cellular functions linked to diseases.

Concerning the section 4 (4.3), a few references should deserve citation and discussion in the disease section, for instance in diabetis or cancer.

-       Elsner M et al. Peroxisome-generated hydrogen peroxide as important mediator of lipotoxicity in insulin-producing cells. Diabetes. 60(1):200-8 (2011)

-       Chen et al. SIRT5 inhibits peroxisomal ACOX1 to prevent oxidative damage and is downregulated in liver cancer. EMBO Rep. 19(5). pii: e45124 (2018).

In addition, I find it regrettable that a small paragraph has not been kept for neurodegenerative diseases, whether or not they are associated with a peroxisomal defect. Oxidative stress is recognized as a major actor of neurodegeneracy in peroxisomal disorders such as X-ALD. It is also true in more common neurodegenerative diseases, especially for hydrogen peroxide. For instance, H2O2 is thought to impact amyloid beta agregation.

Author Response

General comments – The reviewer states: ‘The review is very well written and documented. It provides a very interesting overview of peroxisomal hydrogen peroxide metabolism and signaling. I have however some minor criticisms to formulate.’

We thank the reviewer for his/her thoughtful suggestions. A detailed response to each comment is provided below.

Comment 1 – The reviewer states: ‘Concerning the signaling role of hydrogen peroxide (section 2), I would suggest to mention a few articles dealing with H2O2-regulated transcription factors (NRF2…). The review of Marino et al (Redox Biology 2 :535-62, 2014) may help to build a short paragraph to complete the section. It may help to understand how peroxisomal metabolism can impact various cellular functions linked to diseases.’

As requested by the reviewer, we now include a paragraph illustrating the complexity of H2O2-mediated signaling, with a focus on how this electrophile may modulate the activity of NRF2 (see page 3, lines 108-130). We agree with the reviewer that including such an example may help potential readers to better understand how alterations in peroxisomal metabolism can impact various cellular functions linked to diseases.

Comment 2 – The reviewer states: ‘Concerning the section 4 (4.3), a few references should deserve citation and discussion in the disease section, for instance in diabetis or cancer: Elsner M et al. Peroxisome-generated hydrogen peroxide as important mediator of lipotoxicity in insulin-producing cells. Diabetes. 60(1):200-8 (2011); Chen et al. SIRT5 inhibits peroxisomal ACOX1 to prevent oxidative damage and is downregulated in liver cancer. EMBO Rep. 19(5). pii: e45124 (2018)."

As pointed point in the introductory paragraph of Section 4.3., this section is intended to focus on disease examples that can be linked to alterations in catalase activity. For other examples, we refer to another recent review from our group. However, given that we are fully aware that the manuscript authored by Elsner and colleagues is a key paper in the field, we already cited and discussed the content of this paper in Section 4.2.2. (see lines 275 to 277 in the original version of the manuscript, and lines 301 to 303 in the revised manuscript). Importantly, in deference to the reviewer, we now also cite and discuss the manuscript authored by Chen and co-workers (see page 4, Section 3.1., lines 155 to 161). We hope the reviewer can appreciate our point of view.

Comment 3 – The reviewer states: ‘In addition, I find it regrettable that a small paragraph has not been kept for neurodegenerative diseases, whether or not they are associated with a peroxisomal defect. Oxidative stress is recognized as a major actor of neurodegeneracy in peroxisomal disorders such as X-ALD. It is also true in more common neurodegenerative diseases, especially for hydrogen peroxide. For instance, H2O2 is thought to impact amyloid beta agregation.’

Once again, we agree with the reviewer that oxidative stress is a common denominator of various neurodegenerative disorders, and that virtually all patients suffering from a peroxisomal deficiency typically develop severe neurological deficits. Given that (i) our review specifically focuses on how alterations in peroxisomal H2O2 metabolism and signaling can lead to human disease, and (ii) to the best of our knowledge, little is known
about how peroxisomal H2O2 metabolism contributes to brain homeostasis and health, we didn’t include such a paragraph in the original manuscript. However, given that the reviewer considers this as a gap in our manuscript, we now include a short paragraph (Section 4.3.6.) in which we address this shortcoming (see lines 377 to 397 in the revised manuscript). Note that also in this section we predominantly focus on the link between catalase activity and neurodegenerative disease.